# Abscisic Acid’s Role in the Modulation of Compounds that Contribute to Wine Quality

**DOI:** 10.3390/plants10050938

**Published:** 2021-05-08

**Authors:** Rodrigo Alonso, Federico J. Berli, Ariel Fontana, Patricia Piccoli, Rubén Bottini

**Affiliations:** 1Grupo de Bioquímica Vegetal, Instituto de Biología Agrícola de Mendoza, Facultad de Ciencias Agrarias, Consejo Nacional de Investigaciones Científicas y Técnicas-Universidad Nacional de Cuyo, Almirante Brown 500, 5507 Chacras de Coria, Mendoza, Argentina; realonso@mendoza-conicet.gob.ar (R.A.); afontana@fca.uncu.edu.ar (A.F.); ppiccoli@fca.uncu.edu.ar (P.P.); rbottini48@gmail.com (R.B.); 2Catena Institute of Wine, Bodega Catena Zapata, Cobos s/n, 5509 Agrelo, Mendoza, Argentina; 3Instituto de Veterinaria, Ambiente y Salud, Universidad Juan A. Maza, Lateral Sur del Acceso Este 2245, 5519 Guaymallén, Mendoza, Argentina

**Keywords:** ABA, antioxidant capacity, grapevines, phenolic compounds, UV-B, water deficit

## Abstract

Abscisic acid (ABA) plays a crucial role in the plant responses to environmental signals, in particular by triggering secondary metabolism. High-altitude vineyards in Mendoza, Argentina, are exposed to elevated solar ultraviolet-B (UV-B) levels and moderate water deficits (WD), thus producing grapevine berries with high enological quality for red winemaking. Volatile organic compounds (VOCs) and phenolic compounds (PCs) accumulate in the berry skins, possess antioxidant activity, and are important attributes for red wine. The aim of the present study was to analyze the role of ABA in the modulation of these compounds in *Vitis vinifera* L. cv. Malbec wines by comparing the independent and interactive effects of UV-B, WD, and ABA. Two UV-B treatments (ambient solar UV-B or reduced UV-B), two watering treatments (well-watered or moderate water deficit) and two ABA treatments (no ABA and sprayed ABA) were given in a factorial design during one growing season. Sprayed ABA, alone and/or in combination with UV-B (specially) and WD (to a lower degree) increased low molecular weight polyphenols (LMWP), anthocyanins, but most noticeably the stilbenes *trans*-resveratrol and piceid. Under these treatments, VOCs were scarcely affected, and the antioxidant capacity was influenced by the combination of UV-B and WD. From a technological point of view, ABA applications may be an effective vineyard management tool, considering that it elicited a higher content of compounds beneficial for wine aging, as well compounds related to color.

## 1. Introduction

The plant hormone abscisic acid (ABA) has been proposed as the main mediator of grapevine biological responses to abiotic stresses, including solar ultraviolet-B (UV-B) radiation [1] and water deficit [2]. Also, secondary metabolism was linked to abiotic stress, especially to the ABA-mediated responses [3]. Volatile organic compounds (VOCs) and phenolic compounds (PCs) are secondary metabolites that accumulate in the berry skins, and their biosynthesis are greatly influenced by environmental signals. These compounds are considered the most important attributes for wine quality [4,5], although most scientific works evaluated the interactive effects of environmental factor on these compounds in berries, but not in the wine.

PCs are accumulated in response to various biotic and abiotic stresses, absorbing UV radiation and with antimicrobial and antioxidant properties [6,7]. They include two groups: flavonoids and non-flavonoids; within the flavonoids there are anthocyanins, flavonols (i.e., quercetin and kaempferol), flavanonols (dihydroflavonols, such as astilbin) and flavanols (i.e., catechins, epicatechins, and tannins). Among the non-flavonoids there are stilbenes (such as resveratrol), hydroxycinnamic acids and hydroxybenzoic acids [8]. One of the most important parameters that determine wine quality is its phenolic composition, which is associated with color, flavor, and astringency, but also to bitterness, odor, and oxidative stability of wines [9]. One strategy by which plants can cope with environmental factors is by producing and releasing chemicals such as VOCs [10], which are low molecular weight species including terpenoid (isoprene, monoterpenes and sesquiterpenes) and non-isoprenoid compounds as oxygenated hydrocarbons (alcohols, aldehydes, and ketones) [11]. These compounds may, for instance, protect plants against biotic and abiotic stresses [12,13,14]. VOCs can also include defense mechanisms to control growth of neighboring plants [15]. The aroma of wine is one of the main factors that determines its quality, and derives from both the VOCs present in the berries [16], and those produced during the fermentation and aging [17,18]. Notwithstanding, the role of VOCs coming from berries in wine quality did not received too much attention [19] and most literature has been oriented to study VOC’s as a product of must fermentation [20,21].

In previous reports, we showed that high-altitude vineyards in Mendoza, Argentina, receive relatively high levels of solar UV-B (280–315 nm) that produce berries of cv. Malbec with high enological quality by inducing the synthesis of PCs in the skins, mainly anthocyanins, flavonols, dihydroflavonols, and flavanols [22,23,24]. Likewise, Gil et al. [25] found that UV-B affected the synthesis of VOCs in the berries of cv. Malbec, including monoterpenes, aldehydes, alcohols, and ketones. Additionally, the effects of moderate water deficit irrigation (WD), a common management strategy in viticulture regions with dry climates, has been extensively studied, showing an accumulation of PCs in berries [26]. Nevertheless, there are few studies of the effects of water deficit on the production of VOCs in grapevine berries or wines. Qian et al. [27], found that wine produced from deficit irrigated Merlot vines had increased amounts of vitispiranes, β-damascenone, and guaiacol compounds; however had no effect on the concentrations of other measured volatiles such as esters and terpenes. As well with Merlot, Song et al. [28] observed that deficit irrigation increased the concentration of some non-isoprenoids.

It was observed that wines made with berries from ABA sprayed vineyards had more PCs, mainly anthocyanins [29,30,31], but there are no reports of VOCs profiles in wine from berries of vines treated with ABA. We previously studied the independent and interactive effects of solar UV-B, moderate WD, and ABA applications on cv. Malbec. We found that gas exchange and photosynthesis were reduced by WD and highly impaired in the UV-B and WD combined treatment. Both environmental signals and sprayed ABA elicited mechanisms of acclimation by augmenting in leaves the content of terpenes with antioxidant and antifungal properties [32]. Additionally, solar UV-B increased PCs and antioxidant capacity in berry’s skins, irrespectively of the combination with other factors, while applications of ABA increased total anthocyanins [24].

The aim of the present study was to analyze the ABA role in the modulation of compounds that contribute to the wine quality, by comparing the independent and interactive effects of contrasting UV-B levels, WD and sprayed ABA, on PCs and VOCs profiles of wines.

## 2. Results

### 2.1. LMWP, Anthocyanins and ORAC

Wines from grapevine under +UV-B, regardless of the combination with another factor, showed an increase of 17% (average) in total LMWP (Table 1), with quercetin as the compound that doubled its content. Wines from +WD treated vines presented a higher content of syringic acid (13.2%) and a lower content of quercetin (21.4%) and OH-tyrosol (13.2%), compared with wines from −WD treated vines. Caffeic acid was the other phenolic acid affected, decreasing 18.1% in wines from +ABA treated vines (compared to −ABA treatments). The procyanidins (+)-catechin and (−)-epicatechin, and the prodelphinidin gallocatechin were not affected by the treatments (data not shown). The treatment of +UV-B and +ABA interact on wine’s piceid, i.e., +UV-B/+ABA wines had the highest content of piceid (Figure 1A). Regarding astilbin, UV-B interacted with WD, where the combination of +UV-B and +WD had the highest content as compared with −UV-B (Figure 1B). A triple interaction was observed for the *trans*-resveratrol content, where the +UV-B/+WD/+ABA wines had the highest content showing a synergistic effect of the treatments (6-fold more content than −UV-B/−WD/−ABA and 1.5 times more than +UV-B/+WD/−ABA; Table 1).

Related to anthocyanins, ABA was the main factor that increased their content. Treatments with +ABA, regardless of the combination with another factor, had a total anthocyanin content 30.9% higher than the −ABA wines (Table 2). Analyzing the anthocyanins profile, +ABA wines increased the content of petunidin and malvidin, and also peonidin and delphinidin. In the latter, an interaction between +ABA and +UV-B was also observed (see also Figure 2). The content of cyanidin in +UV-B wines was 4.8% lower than in −UV-B wines.

Glycosylated (non-acylated) anthocyanins were the most abundant anthocyanins in the wines of all treatments (Table 3). Treatments with +ABA shown a higher content of both acetylated and p-coumaroylated forms, while non-acylated forms increased by +ABA and also by the combination of +ABA/+UV-B (significant interaction; Figure 2C).

### 2.2. VOCs

Twenty-six compounds were identified in the volatile fraction of Malbec wines. Among them: two aliphatic ketones, four aliphatic alcohols, one aromatic alcohol, three aliphatic acids, 11 esters aminoacid glycine were detected (Table 4, Table 5 and Table 6). Some compounds were affected independently by one factor, while double and triple interactions between factors were also observed. The ethyl esters of octanoate and decanoate were the most abundant compounds, while 2-phenylethanol was the alcohol more representative (Table 4). Generally, the concentration of total volatile alcohols was diminished by +UV-B, +D and +ABA, but noticeably when the stress factors were combined. It is seen in Table 4 that isoprenoids and non-isoprenoid alcohols such iso-butanol and nerolidol decreased 46.2% and 22.7%, respectively, due to +UV-B, while hexanol decreased 29.8% due to +ABA. The wines obtained from berries under +WD had a lower content of alcohols citronellol (18.2%), 2-phenylethanol (31.3%) and nerolidol (37.7%), compared to −WD wines. The 3-methyl-1-pentanol was affected by the combination of UV-B and WD, where in −UV-B/+WD wine the lowest content was found (significant interaction; Figure 3A). From the non-isoprenoid alcohols, the most important ones found in wines, 2,3 butanediol and 1,3 butanediol were affected by WD and ABA, being higher in −WD/−ABA wines and lower in the rest of the combinations (Figure 3B,C). While the terpene linalool had the highest content in −UV-B/−ABA wines (Figure 3D).

The content of ethyl decanoate and ethyl pentadecanoate were reduced only by +WD, 30.7% and 67.7%, respectively, compared to −WD wines, while ethyl nonanoate decreased 28.4% due to +ABA (Table 5). The esters isoamyl acetate, ethyl octanoate, octyl formate and ethyl palmitate were not affected by the treatments (data not shown). The octanoic acid was reduced in +WD by 47.5% (Table 6), and hexanoic acid was affected by combined stressful factors (Figure 4D). The total volatile esters were mostly reduced by UV-B and WD, but especially in combined +UV-B/+WD (Table 5; Figure 5A). Respect to ORAC, the −WD/+UV-B wines presented a higher ORAC as compared to −WD/−UV-B wines (Figure 5B).

## 3. Discussion

In a previous report, we observed that PCs increased by +UV-B treatments in the cv. Malbec berry skin extracts from the high-altitude vineyard, but the differences disappear after the winemaking process as assessed by spectrophotometric analysis of the wines [33]. In the present work, using more precise analytical techniques for the wine, we observed that the content of LMWP was higher (17%) in +UV-B wines, being quercetin the compound that contributed most to this increase. As well, +UV-B wines showed the LMWP with highest antioxidant capacity that have been previously observed in berry skins [24]. Like in berries, astilbin was also increased in +UV-B wines, although in combination with +WD. Respect to antioxidant capacity, it was observed that only the +UV-B/−WD wines presented the highest values, measuring by the ORAC technique. According to Rodríguez-Bonilla et al. [34], the use of other complementary technique would be necessary, since we observed that UV-B increased the LMWP with highest antioxidants capacity, regardless of the water status.

In Alonso et al. [24] we found that fruit yield was reduced in +UV-B/+WD and in +ABA/+WD treatments, i.e., WD only affected yield in plants under high intensity of UV-B or in combination with ABA applications. In addition, ABA sprays may anticipate the accumulation of sugars in berries, but major effects of ABA are found at veraison and differences are reduced at harvest [21]. Wines +WD had a lower content of quercetin, in correlation with what happened in the skins. Possibly, WD increased in the skins the expression of the enzyme flavonoid 3′5′ hydroxylase, which competes for the same substrate with the enzyme flavonoid 3-hydroxylase that mediates quercetin biosynthesis [35]. Syringic acid has been indicated as one of the hydroxycinnamic acids with the greatest co-pigmentation effect with malvidin [36], which improves color stability of the wine.

Like in skins [24], it was observed that the (+)-catechin and (−)-epicatechin contents in the wine were not affected by the treatments. Deis et al. [29] found in cv. Cabernet Sauvignon that post-veraison WD (stem water potentials between −1.0 and −0.5 MPa, a moderate intensity according to Van Leeuwen et al. [37]) and applied ABA (doses of 250 mg L^−1^) increased (+)-catechin in berries and wines, while Koundouras et al. [38] did not observe an effect of post-veraison WD (stem water potentials between −1.2 and −0.8 MPa) in cv. Agiorgitiko berries or wine.

The biosynthesis of stilbenes, including *trans*-resveratrol, is increased in berries by environmental signals such as WD (stem water potentials between −1.25 and −0.8 MPa, [39]) and UV-B [33], and also by ABA applications [29,40]. In the present work, the wine of those plants that received both environmental signals and ABA applications had 6-fold more *trans*-resveratrol as compared to the treatment −UV-B/-D/−ABA. The stilbene piceid (also called polydatin or resveratrol-3-O-b-D-glucoside) is a precursor of resveratrol with a capacity to trap hydroxyl radicals greater than vitamin C [41] and also with cardioprotective activity [42]. Sprayed ABA to vines submitted to relatively high UV-B irradiances (+UV-B/+ABA) resulted in wines with a high content of piceid.

Former studies have shown that ABA applications increased the anthocyanin content in the wine obtained, although the times of application and the concentrations used were different from the present work [30,31,40]. Quiroga et al. [40] applied 250 ppm of ABA weekly, from veraison to harvest, while Xi et al. [30] and Luan et al. [31] sprayed 200 ppm of ABA at pre-veraison. In the present study, we show that with only two applications (in veraison and 15 days later) the total anthocyanin content increased. Wines from +ABA treated vines, regardless of the combination with another factor, increased the trihydroxylated anthocyanins petunidin and malvidin, which are indicated in the literature as the most oxidation-stable anthocyanins [43]. The latter is important from the oenological point of view since the anthocyanins that increased are compounds that would keep the color stable during aging. An interaction effect of +UV-B/+ABA was also observed in wines. Possibly ABA applications increased in berries the expression of UDP-glucose flavonoid glucosyl transferase (UFGT) responsible for the glycosylation of anthocyanins [44]. Additionally, Carbonell-Bejerano et al. [45] observed that UV-B increased the expression of the UFGT enzyme in berries of cv. Tempranillo.

Although ABA applications increased certain compounds regardless of the intensity of UV-B received, the greater effectiveness of +ABA was observed in plants under high UV-B irradiances, since those compounds with high antioxidant capacity, such as piceid, and those related to oenological quality, such as peonidin and delphinidin, were markedly increased (as compared to −UV-B treatment). Similar results were observed in a previous work, where ABA applications in plants under +UV-B, produced the highest berry skin ABA levels, increasing additively flavonols and anthocyanins [22]. In general, the levels of VOCs in wines decreased due to stressful factors, and ABA sprays had a lower effect. The present is the first work that evaluates the effect of water restriction and ABA applications on the profile of VOCs in wines of cv. Malbec. Among the non-isoprenoid alcohols, the isobutanol is one of the volatiles that decreased more (46.2%) in response to the incidence of +UV-B. The most abundant alcohols in wine, including isobutanol, are produced by secondary reactions of yeasts during fermentation, either through anabolic pathways from glucose or through catalytic reactions from amino acids [46]. Martinez-Luscher et al. [47] observed a decrease in the contents of the aminoacid isoleucine (precursor of isobutanol) due to the effect of +UV-B. Purportedly, UV-B decreased the isoleucine contents in the berries, and this would explain the lower isobutanol content that we observed in the wine, although more studies will be necessary to confirm this hypothesis. Isobutanol in wines is associated with an herbaceous aroma, and Ribéreau-Gayon et al. [48] mentioned that low concentrations of this alcohol contribute to a greater aromatic complexity of the wine. Phenethyl alcohol (also called 2-phenylethanol) is an aromatic alcohol and considered one of the most abundant in wine after ethanol [49], mainly produced during the fermentation process from the aminoacid phenylalanine [50]. In the present study, the concentration of 2-phenylethanol in the wines decreased only by +WD, while Del-Castillo-Alonso et al. [51] found that supplemented UV-B moderately decreased the contents in Tempranillo wines. Hexanol is one of the C6 aliphatic compounds associated with a vegetal character in the wine. Although it is present in berries, most of them derive from the degradation of polyunsaturated fatty acids, via lipoxygenase, when cell membranes are broken during grape grinding [52]. A decrease of almost 30% in hexanol content was observed in wines of +ABA treated vines. The alcohols 2, 3 butanediol and 1, 3 butanediol, whose descriptors in wine are butter aromas [53], were affected by WD and ABA. These alcohols come mostly as secondary products of the alcoholic fermentation carried out by yeasts [48]. Respect to the non-isoprenoid short-medium chain organic acids, the aliphatic hexanoic and octanoic acids are included; both acids are produced by yeasts during alcoholic fermentation [54]. Reports from Tao and Zhang [55] and Jiang et al. [56] show their responsibility of the pleasant odor at low concentrations. On the other hand, fatty acid esters in wine are also compounds derived from the metabolism of yeasts, and may contribute to the fruity notes of the wine [46]. The +ABA wines had a lower content of ethyl nonanoate, while +WD decreased the ethyl decanoate and ethyl pentadecanoate.

Malbec belongs to the group of varieties with very low concentration of terpenoids and possibly explain the discrepancies of our results with those of Ou et al. [57], whose midday leaf water potentials of WD treatments were between −1.09 and −1.63 (moderate to severe intensity, [37]). In the present work, we only identified four terpenic alcohols, where linalool led the highest content in wines from −UV-B/−ABA combination, while WD had no effect. Ou et al. [57] also observed that the linalool content did not vary in wines from plants of cv. Merlot under WD, although the citronellol content increased by restricted irrigation. It is known that terpenes as linalool, citronellol, nerolidol and farnesol are present in grapes of Malbec [58], being farnesol the precursor of many sesquiterpenes associated with floral notes, found in much higher concentration in Muscat varieties and Gewurztraminer [59] than in red varieties such as Malbec.

## 4. Materials and Methods

### 4.1. Plant Material and Experimental Design

The experiment was conducted during the 2012–2013 growing season using *Vitis vinifera* L. cv. Malbec vines grown in a high-altitude vineyard of 15 years old in Gualtallary, Mendoza, Argentina (69°15′37″ W; 33°23′51″ S) at 1450 m a.s.l., as it is described in [32]. Briefly, a low UV-B treatment (−UV-B) was set by using a polyester cover that absorbed 78% of UV-B and 18% of ultraviolet-A (UV-A) radiation; and a close to ambient UV-B treatment (+UV-B) was established with a low-density polyethylene that transmitted 90% of UV-B and 87% of UV-A. The UV-B treatments were given from 15 days after flowering (DAF) until harvest at 142 DAF. Vines were maintained with no soil water restriction until veraison (84 DAF), and then two irrigation regimes were set, a well-watered treatment (−WD) and a moderate water deficit treatment (+WD), maintaining stem water potentials at midday of −0.7 and −1.0 MPa, respectively. Additionally, the aerial part of plants was sprayed at veraison and repeated once 15 days after, with 1 mM ABA (+ABA; ±-cis, trans-abscisic acid, Kelinon Agrochemical Co., Beijing, China) or water (−ABA) solutions containing 0.1% v/v of Triton X-100 as surfactant.

A subdivided plot design was used, and the main plot corresponded to factor WD, whose levels (+WD and −WD) were arranged in a completely randomized 5-block design. The 10 main parcels were divided up into two sub-parcels to which the levels of the UV-B factor were randomly assigned, and each sub-parcel was again divided into two and assigned to the random levels of factor ABA. The experimental unit consisted of two plants (selected based on their homogeneity) from six consecutive plants in the row. In summary, a total of eight combined treatments were performed: (i) +UV-B/+WD/+ABA; (ii) +UV-B/−WD/+ABA); (iii) +UV-B/+WD/−ABA; (iv) +UV-B/−WD/−ABA; (v) −UV-B/+WD/+ABA; (vi) −UV-B/−WD/+ABA; (vii) −UV-B/+WD/−ABA; and (viii) −UV-B/−WD/−ABA.

### 4.2. Winemaking

The grapes were harvested at 142 DAF, coinciding with the commercial harvest time of the plot (about 25° Brix). Microfermentations were performed to obtain three replicates for each combined treatment. The berries were separated manually from the clusters, weighing ca. 1.4 kg per replicate, and placed in a plastic vessel. The grapes were crushed and inoculated with 200 mg/kg of selected commercial *Saccharomyces cerevisiae bayanus* yeast. During fermentation the temperature was kept at 25 °C and density was measured daily. When the alcoholic fermentation was completed, the solid parts were separated and potassium metabisulfite (200 mg) was added. The temperature was maintained at 5 °C for one week, and then 750 mL of the upper fraction were bottled and kept at room temperature. Additionally, 15 mL of the upper fraction were placed in conical tubes and kept at −20 °C.

### 4.3. Analysis of LMWP, Anthocyanins and ORAC in Wine

LMWPs and anthocyanins were assessed in August 2013 using a high performance liquid chromatography-multiple wavelength detector (HPLC-MWD; Dionex Ultimate 3000, Dionex Softron GmbH, Thermo Fisher Scientific Inc., Germering, Germany) following the procedures described by Fontana et al. [60]. For LMWPs, a 5 mL aliquot of wine was acidified with 57 μL of 1% FA. Then 2.5 mL of MeCN was added and it was stirred vigorously for 60 sec. Later, 4 g of Na_2_SO_4_ and 1.5 g of NaCl were added. The mixture was stirred again for 60 sec and centrifuged for 10 min at 300 rpm. One ml was extracted from the MeCN phase and 150 mg of CaCl_2_, 100 mg of PSA and 100 mg of C_18_ were added. The mixture was vortexed for 30 sec and centrifuged 2.5 min at 12,000 rpm. Finally, a 500 μL aliquot of the supernatant was transferred to a vial and evaporated to dryness with a stream of N_2_. The residue was resuspended with 500 μL of the initial mobile phase and injected into the HPLC-MWD. HPLC separations were carried out in reversed-phase Kinetex C_18_ column (3.0 mm × 100 mm, 2.6 µm) Phenomenex (Torrance, CA, USA). Ultrapure water with 0.1% FA (A) and MeCN (B) were used as mobile phases. Analytes were separated using the following gradient: 0–2.7 min, 5% B; 2.7–11 min, 30% B; 11–14 min, 95% B; 14–15.5 min, 95% B; 15.5–17 min, 5% B: 17–20, 5% B. The mobile phase flow was 0.8 mL min^−1^. The column temperature was 35 °C and the injection volume 5 μL. The identification and quantification were carried out by comparing the retention time and the maximum absorbance value detected in a peak of the sample with those obtained in the injection of the pure standards.

For anthocyanins, HPLC separations were carried out in reversed-phase C18 column (3.0 mm × 100 mm, 2.6 µm, Kinetex, Phenomenex; Torrance, CA, USA). The mobile phase consisted of ultra-pure water/ FA/ MeCN (87: 10: 3, v/v/v; eluent A) and ultra-pure water/FA/MeCN (40:10:50, v/v/v; eluent B), using the following gradient: 0 min, 10% B; 0–6 min, 25% B; 6–10 min, 31% B; 10–11min, 40% B; 11–14 min, 50% B; 14–15 min, 100% B; 15–17 min, 10% B; 17–21 min, 10% B. The mobile phase flow was 1 mL min^−1^, the column temperature was kept at 25 °C and the injection volume was 5 μL. Quantification was carried out by measuring the area under the peaks at 520 nm, and the anthocyanin content was expressed as malvidin-3-glucoside, using a calibration curve with an external standard (1–250 µg mL^−1^, R^2^ = 0.9984).

The Chromeleon 7.1 software was used to control all the acquisition parameters of the LC-MWD system and also to process the obtained data.

The oxygen radical absorbance capacity (ORAC) was determined according to Berli et al. [23] with a microplate fluorometer (Fluoroskan Ascent FL; Thermo Fisher Scientific, Wilmington, USA), and expressed as μmol Trolox equivalent per ml of wine.

### 4.4. Analysis of Wine VOCs

The wine samples were analyzed in August 2013 by headspace solid-phase microextraction (HS-SPME), following the procedures described by Xi et al. [61] with some modifications: 8 mL of wine and 20 μL of 2-octanol (1:2000; internal standard) were transferred into a 20 mL vial, and 1.5 g of NaCl added. The vial was sealed with a Teflon septum and equilibrated at 40 °C for 10 min, shaken by a magnetic stirrer (540 rpm). The fiber PDMS (100 mm polydimethylsiloxane, Supelco, Bellefonte, PA, USA) was inserted into the headspace, where extraction was allowed to occur for 30 min with continued heating (40 °C) and shaking. This was immediately followed by desorption of the analytes into the gas chromatography-electron impact mass spectrometer (GC-MS; Clarus 500, PerkinElmer, Shelton, CT, USA). Separation of compounds was performed with an Agilent DB-Wax capillary column (30 m × 0.32 mm, 0.25 μm) using helium as carrier gas at a flow rate of 1 mL min^−1^. The oven temperature program was as follows: 40 °C for 3 min, then it increased to 160 °C at 5 °C min^−1^, then it was raised to 230 °C at 7 °C min^−1^ and finally it was maintained 8 min at 230 °C. The mass spectrometer was operated using an impact energy of 70 eV and scanned in a range of 35 to 200 atomic mass units. The metabolites were identified by comparison of GC retention times and full mass spectra of the corresponding standard obtained by Fluka (Sigma-Aldrich, Steinheim, Switzerland) and, by using their fragmentation pattern and comparison with data of the NIST library. Peak areas were referred to the standard 2-octanol for quantification.

### 4.5. Standards

Standards of gallic acid (99%), 3-hydroxytyrosol (99.5%), (−)- gallocatechin (≥98%), caftaric acid (≥97%), (−)-epigallocatechin (≥95%), (+)-catechin (≥99%), (−)-epicatechin (≥95%), (−)- epigallocatechin gallate (≥95%), caffeic acid (99%), syringic acid (≥95%), coumaric acid (99%), ferulic acid (≥99%), piceatannol (99%), trans-resveratrol (≥99%), quercetin hydrate (95%), cinnamic acid (99%), quercetin 3-b-D-glucoside (≥90%), kaempferol-3-glucoside (≥99%) and malvidin-3-O- glucoside chloride (≥95%) were purchased from Sigma–Aldrich. The standard of 2-(4-hydroxyphenyl) ethanol (tyrosol) (≥99.5%) was obtained from Fluka (Buchs, Switzerland). Stock solutions of the above mentioned compounds were prepared in methanol at concentration levels of 1000 µgm l^−1^. Calibration standards were dissolved in the initial mobile phase of each method (LMWP or anthocyanins, respectively). HPLC-grade Acetonitrile (MeCN) and formic acid (FA) were acquired from Mallinckrodt Baker (Inc. Phillipsburg, NJ, USA). Primary-secondary amine (PSA) and octa- decylsilane (C_18_) were both obtained from Waters (Milford, MA, USA). Reagent grade NaCl, anhydrous Na_2_CO_3_, anhydrous MgSO_4_ and anhydrous CaCl_2_ were purchased from Sigma–Aldrich. Ultrapure water was obtained from a Milli-Q system (Millipore, Billerica, MA, USA).

### 4.6. Statistical Analysis

The effects of UV-B (*P_UV-B_*), ABA (*P_ABA_*), water restriction (*P_WD_*) and their interactions were determined with three-way ANOVA for a subdivided plots design and the differences between the means were analyzed with Fisher’s LSD test (*p* ≤ 0.05) using InfoStat software (version 2019, Grupo InfoStat, Córdoba, Argentina).

## 5. Conclusions

Ambient +UV-B, regardless of the combination with another factor, increased total LMWP (Quercetin doubled) in the wine. +UV-B/+ABA wines had the highest content of piceid, precursor of *trans*-resveratrol, and the content of the latter was augmented 6-fold by +UV-B/+WD/+ABA as compared to −UV-B/−WD/−ABA. ABA increased anthocyanins, free as well as conjugated, alone or in combination with +UV-B. Under these treatments, VOCs were moderately affected. From a technological point of view, ABA applications may be an effective vineyard management tool, considering the wines obtained had a higher content of antioxidant compounds (beneficial for aging), and of those compounds related to color (quality parameter). The effectiveness of ABA applications is scarcely affected by WD, but noticeably improved in vines exposed to high levels of UV-B.

## Figures and Tables

**Figure 1 plants-10-00938-f001:**
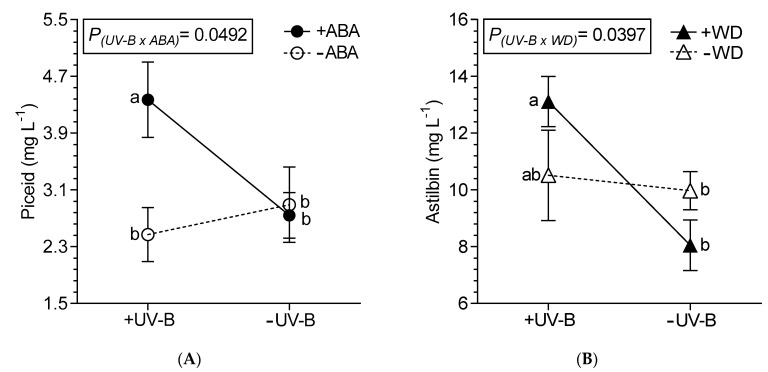
Interaction between UV-B and ABA for piceid content (**A**) and between UV-B and WD for astilbin content (**B**) determined in wines from Malbec plants exposed to combined UV-B, WD and ABA treatments. Values are means for each factor combination ± SEM (n = 3) and different letters indicate significant differences (Fisher’s LSD, *p* ≤ 0.05).

**Figure 2 plants-10-00938-f002:**
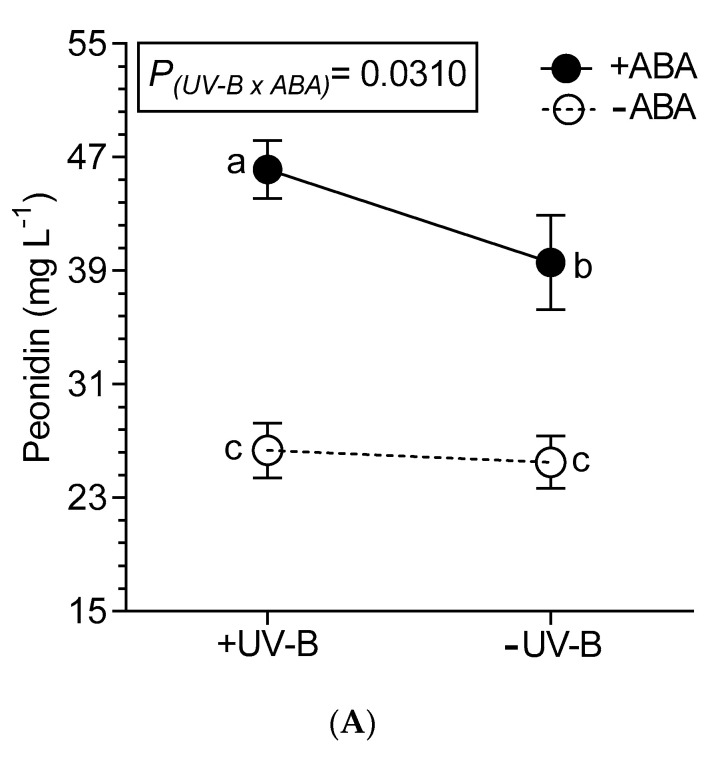
Interaction between UV-B and ABA for peonidin content (**A**), delphinidin content (**B**) and total non-acylated anthocyanins (**C**) determined in wines from Malbec plants exposed to combined UV-B, WD and ABA treatments. Values are means for each factor combination ±SEM (n = 3) and different letters indicate significant differences (Fisher’s LSD, *p* ≤ 0.05).

**Figure 3 plants-10-00938-f003:**
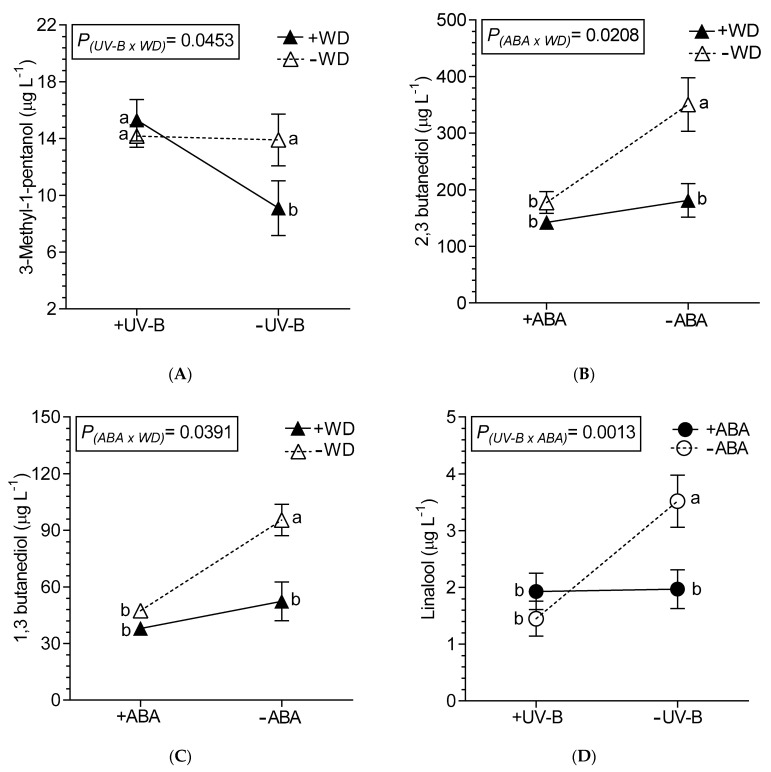
Interaction between UV-B and WD for 3-methyl-1-pentanol (**A**), between ABA and WD for 2,3 butanediol (**B**) and 1,3 butanediol (**C**), and between UV-B and ABA for linalool (**D**), determined in wines from Malbec plants exposed to combined UV-B, WD and ABA treatments. Values are means for each factor combination ±SEM (n = 3) and different letters indicate significant differences (Fisher’s LSD, *p* ≤ 0.05).

**Figure 4 plants-10-00938-f004:**
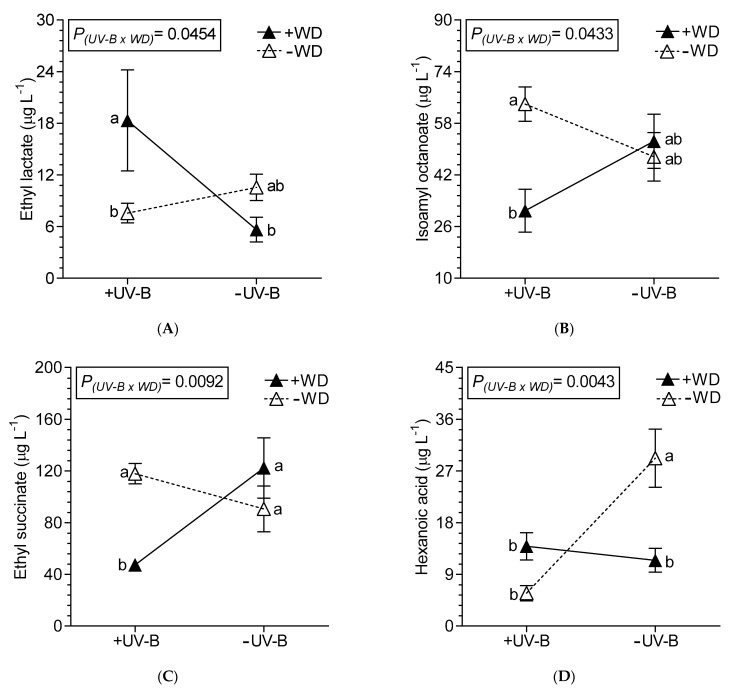
Interaction between UV-B and WD for ethyl lactate (**A**), isoamyl octanoate (**B**), ethyl succinate (**C**) and hexanoic acid (**D**), determined in wines from Malbec plants exposed to combined UV-B, WD and ABA treatments. Values are means for each factor combination ±SEM (n = 3) and different letters indicate significant differences (Fisher’s LSD, *p* ≤ 0.05).

**Figure 5 plants-10-00938-f005:**
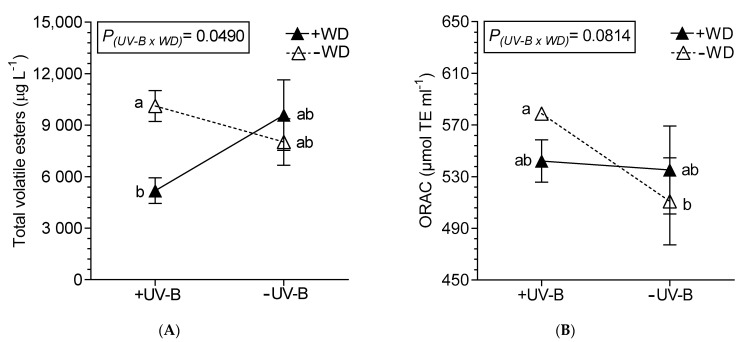
Interaction between UV-B and WD for total volatile esters (**A**) and ORAC (**B**), determined in wines from Malbec plants exposed to combined UV-B, WD and ABA treatments. Values are means for each factor combination ±SEM (n = 3) and different letters indicate significant differences (Fisher’s LSD, *p* ≤ 0.05).

**Table 1 plants-10-00938-t001:** Low molecular weight phenolic (LMWP; mg L^−1^) determined in wines from Malbec plants exposed to combined UV-B, WD and ABA treatments.

Treatments	Syringic Acid	Cafeic Acid	Piceid	Resveratrol	Astilbin	Quercetin	OH-tyrosol	Total LMWP
**UV-B**								
+UV-B	6.1 ± 0.21	1.44 ± 0.10	3.54 ± 0.42	1.85 ± 0.30	11.90 ± 1.00a	5.82 ± 0.65a	3.47 ± 0.19	111.2 ± 4.6a
−UV-B	6.8 ± 0.23	1.58 ± 0.14	2.82 ± 0.29	1.15 ± 0.23	9.01 ± 0.61b	2.79 ± 0.31b	3.03 ± 0.21	94.9 ± 3.5b
**ABA**								
+ABA	6.4 ± 0.20	1.36 ± 0.11b	3.67 ± 0.38	1.98 ± 0.30a	9.81 ± 0.90	4.47 ± 0.61	3.17 ± 0.25	103.2 ± 6.0
−ABA	6.5 ± 0.27	1.66 ± 0.12a	2.68 ± 0.32	1.02 ± 0.18b	11.1 ± 0.84	4.14 ± 0.72	3.32 ± 0.17	100.9 ± 2.9
**WD**								
+WD	6.8 ± 0.28a	1.43 ± 0.10	3.09 ± 0.32	1.94 ± 0.32	10.67 ± 1.00	3.79 ± 0.67b	3.02 ± 0.21b	103.5 ± 6.2
−WD	6.0 ± 0.11b	1.58 ± 0.14	3.26 ± 0.40	1.06 ± 0.16	10.24 ± 0.83	4.82 ± 0.62a	3.48 ± 0.19a	101.2 ± 2.6
**+UV-B**								
+WD/+ABA	6.8 ± 0.88	1.45 ± 0.09	4.53 ± 0.54	3.43 ± 0.24a	13.68 ± 0.85	6.42 ± 1.90	3.53 ± 0.65	134.9 ± 1.9
−WD/+ABA	5.9 ± 0.03	1.24 ± 0.19	4.27 ± 0.91	1.63 ± 0.25b	8.66 ± 1.03	5.39 ± 0.70	3.43 ± 0.41	101.3 ± 5.0
+WD/−ABA	6.2 ± 0.42	1.42 ± 0.10	2.36 ± 0.47	1.38 ± 0.36b	12.73 ± 1.48	4.67 ± 1.03	3.28 ± 0.10	103.7 ± 4.8
−WD/−ABA	5.6 ± 0.19	1.54 ± 0.34	2.59 ± 0.70	0.92 ± 0.29bc	12.36 ± 2.85	6.71 ± 1.93	3.7 ± 0.54	113.5 ± 0.7
**−UV-B**								
+WD/+ABA	6.7 ± 0.53	1.05 ± 0.11	2.42 ± 0.30	1.66 ± 0.63b	7.27 ± 1.77	2.1 ± 0.42	2.25 ± 0.36	93.8 ± 14
−WD/+ABA	6.3 ± 0.13	1.59 ± 0.29	3.07 ± 0.55	1.16 ± 0.21bc	9.47 ± 1.35	3.89 ± 0.62	3.54 ± 0.32	95.3 ± 1.0
+WD/−ABA	7.7 ± 0.41	1.72 ± 0.22	2.65 ± 0.51	1.24 ± 0.46bc	8.83 ± 0.52	1.9 ± 0.20	3.06 ± 0.27	91.9 ± 3.1
−WD/−ABA	6.3 ± 0.25	1.96 ± 0.19	3.13 ± 1.04	0.49 ± 0.03c	10.47 ± 0.41	3.28 ± 0.44	3.26 ± 0.43	98.5 ± 5.1
**ANOVA ^a^**								
*p_(UV-B)_*	0.0879	0.1093	0.2691	0.2774	0.1732	**0.0133**	0.2502	**0.0474**
*p_(ABA)_*	0.5959	**0.0388**	0.1115	**0.0022**	0.2672	0.8516	0.674	0.6436
*p_(WD)_*	**0.0482**	0.3188	0.5782	0.0926	0.7815	**0.0433**	**0.0205**	0.4717
*p_(UV-B ×ABA)_*	0.2134	0.2932	**0.0492**	0.0551	0.877	0.8735	0.7796	0.3217
*p_(UV-B × WD)_*	0.6836	0.2039	0.7275	0.6226	**0.0397**	0.2806	0.0776	0.2315
*p_(ABA ×W D)_*	0.2417	0.8103	0.9083	0.2837	0.5167	0.5656	0.6236	0.1341
*p_(UV-B ×ABA × WD)_*	0.3757	0.3610	0.7286	**0.0392**	0.3585	0.4096	0.2731	0.1615

Values are means for each factor or treatment (n = 3) ± SEM. Different lowercase letters within columns indicate significant differences (Fisher’s LSD, *p* ≤ 0.05). ^a^ *p*_(UV-B)_, *p*_(ABA)_ and *p*_(WD)_: effects of UV-B, ABA and WD, respectively; *p*_(UV-B ×ABA)_, *p*_(UV-B ×W D)_, *p*_(ABA × WD)_, and *p*_(UV-B ×ABA × WD)_: interaction effects of factors. *p*-values ≤ 0.05 were considered statistically significant, and are in bold.

**Table 2 plants-10-00938-t002:** Anthocyanin content (mg L^−1^) determined in wines from Malbec plants exposed to combined UV-B, WD and ABA treatments.

Treatments	Dihydroxylated		Trihydroxylated			Total Antho
	Cyanidin	Peonidin	Delphinidin	Petunidin	Malvidin	
**UV-B**						
+UV-B	8.67 ± 0.14b	36.2 ± 3.3	50.7 ± 4.6	71.0 ± 5.2	302.3 ± 14.0	468.8 ± 26.6
−UV-B	9.11 ± 0.19a	33.3 ± 2.9	44.1 ± 4.8	65.3 ± 5.6	292.1 ± 12.0	444.0 ± 24.8
**ABA**						
+ABA	8.85 ± 0.15	42.8 ± 2.1	58.2 ± 3.6	80.8 ± 3.8a	326.8 ± 9.0a	517.4 ± 17.7a
−ABA	8.93 ± 0.20	26.7 ± 1.3	36.6 ± 2.8	55.6 ± 3.3b	267.7 ± 9.3b	395.4 ± 15.8b
**WD**						
+WD	8.63 ± 0.13	32.5 ± 3.1	43.1 ± 4.8	62.8 ± 5.5	288.5 ± 14.4	435.5 ± 27.1
−WD	9.16 ± 0.19	37.1 ± 3.1	51.6 ± 4.4	73.5 ± 4.9	305.9 ± 11.1	477.3 ± 22.9
**+UV-B**						
+WD/+ABA	8.36 ± 0.12	46.6 ± 3.3	66.2 ± 3.2	88.0 ± 4.8	347.7 ± 19.5	556.9 ± 30.6
−WD/+ABA	8.71 ± 0.16	45.7 ± 3.2	60.8 ± 5.8	83.1 ± 5.4	334.5 ± 8.1	532.8 ± 19.8
+WD/−ABA	8.81 ± 0.44	24.8 ± 2.1	34.4 ± 4.3	52.0 ± 3.6	252.6 ± 9.3	372.6 ± 13.8
−WD/−ABA	8.79 ± 0.33	27.9 ± 3.5	41.3 ± 7.1	60.9 ± 8.7	274.2 ± 24.8	413.0 ± 43.6
**−UV-B**						
+WD/+ABA	8.80 ± 0.18	34.8 ± 4.0	44.4 ± 5.2	65.8 ± 7.3	295.7 ± 23.2	449.5 ± 39.5
−WD/+ABA	9.54 ± 0.23	44.4 ± 4.0	61.6 ± 8.4	86.2 ± 7.7	329.0 ± 6.8	530.7 ± 26.3
+WD/−ABA	8.54 ± 0.26	23.7 ± 2.6	27.6 ± 4.4	45.4 ± 6.0	258.0 ± 26.0	363.2 ± 38.8
−WD/−ABA	9.52 ± 0.73	28.1 ± 1.3	38.3 ± 1.5	58.8 ± 2.6	274.8 ± 9.6	409.7 ± 15.6
**ANOVA ^a^**						
*p_(UV-B)_*	**0.0175**	0.2748	0.1379	0.2837	0.5404	0.3797
*p_(ABA)_*	0.7196	**<0.0001**	**<0.0001**	**<0.0001**	**0.0005**	**<0.0001**
*p_(WD)_*	0.1341	0.1248	0.0911	0.0527	0.1480	0.0777
*p_(UV-B ×ABA)_*	0.3008	**0.0310**	**0.0428**	0.0668	0.1254	0.0610
*p_(UV-B × WD)_*	0.2479	0.1779	0.0973	0.0742	0.2080	0.1079
*p_(ABA ×WD)_*	0.7943	0.6731	0.0943	0.0937	0.5021	0.2967
*p_(UV-B ×ABA × WD)_*	0.4382	0.3299	0.2246	0.1555	0.3103	0.2238

Values are means for each factor or treatment (n = 3) ± SEM. Different lowercase letters within columns indicate significant differences (Fisher’s LSD, *p* ≤ 0.05). ^a^
*p*_(UV-B)_, *p*_(ABA)_ and *p*_(WD)_: effects of UV-B, ABA and WD, respectively; *p*_(UV-B ×ABA)_, *p*_(UV-B × WD)_, *p*_(ABA × WD)_, and *p*_(UV-B ×ABA × WD)_: interaction effects of factors. *p*-values ≤ 0.05 were considered statistically significant, and are in bold.

**Table 3 plants-10-00938-t003:** Anthocyanins (grouped by acylation; mg L^−1^) determined in wines from Malbec plants exposed to combined UV-B, WD and ABA treatments.

Treatments	Non-Acylated	Acetylated	p-Coumarylated
**UV-B**			
+UV-B	349.6 ± 20.3	62.0 ± 2.6	57.3 ± 3.9
−UV-B	327.8 ± 19.0	59.7 ± 2.8	56.5 ± 3.5
**ABA**			
+ABA	384.9 ± 14.2a	67.1 ± 1.7a	65.4 ± 2.2a
−ABA	292.4 ± 11.8b	54.6 ± 1.6b	48.4 ± 2.7b
**WD**			
+WD	324.1 ± 20.8	58.5 ± 2.6	53.0 ± 3.8
−WD	353.3 ± 17.8	63.2 ± 2.3	60.8 ± 3.2
**+UV-B**			
+WD/+ABA	418.3 ± 21.9	70.1 ± 2.7	68.5 ± 6.2
−WD/+ABA	396.6 ± 18.4	68.3 ± 1.8	67.9 ± 2.3
+WD/−ABA	277.0 ± 11.7	52.1 ± 1.6	43.5 ± 1.3
−WD/−ABA	306.4 ± 32.3	57.4 ± 4.3	49.2 ± 7.1
**−UV-B**			
+WD/+ABA	331.7 ± 31.4	60.7 ± 4.0	57.1 ± 4.3
−WD/+ABA	393.2 ± 23.6	69.3 ± 2.8	68.1 ± 1.2
+WD/−ABA	269.3 ± 29.6	51.0 ± 3.1	42.9 ± 6.3
−WD/−ABA	298.1 ± 11.3	56.1 ± 1.2	55.5 ± 3.1
**ANOVA ^a^**			
*p_(UV-B)_*	0.3303	0.5185	0.7290
*p_(ABA)_*	**<0.0001**	**0.0001**	**0.0004**
*p_(WD)_*	0.1039	0.0507	0.0676
*p_(UV-B ×ABA)_*	**0.0443**	0.1901	0.1768
*p_(UV-B × WD)_*	0.1209	0.1192	0.1418
*p_(ABA × WD)_*	0.2808	0.2823	0.4665
*p_(UV-B ×ABA × WD)_*	0.1670	0.2357	0.7077

Values are means for each factor or treatment (n = 3) ± SEM. Different lowercase letters within columns indicate significant differences (Fisher’s LSD, *p* ≤ 0.05). ^a^
*p*_(UV-B)_, *p*_(ABA)_ and *p*_(WD)_: effects of UV-B, ABA and WD, respectively; *p*_(UV-B ×ABA)_, *p*_(UV-B × WD)_, *p*_(ABA × WD)_, and *p*_(UV-B ×ABA × WD)_: interaction effects of factors. *p*-values ≤ 0.05 were considered statistically significant, and are in bold.

**Table 4 plants-10-00938-t004:** Volatile alcohols (µg L^−1^) determined in wines from Malbec plants exposed to combined UV-B, WD and ABA treatments.

Treatmens	Iso-butanol	3-Methyl-1-pentanol	Hexanol	2,3 butanediol	Linalool	1,3 butanediol	Citronellol	2-Phenylethanol	Nerolidol	Glycerin	Total Alcohols
**UV-B**											
+UV-B	170.5 ± 29.6b	14.7 ± 0.8	66.3 ± 8.2	185.9 ± 18.7b	1.7 ± 0.2	55.2 ± 6.7	16.3 ± 1.3	2154.4 ± 261.8	8.5 ± 1.1b	27.6 ± 10.4	2707.8 ± 298.3b
−UV-B	316.7 ± 46.2a	11.5 ± 1.5	57.6 ± 5.8	240.0 ± 39.1a	2.7 ± 0.4	61.4 ± 9.3	19.0 ± 1.9	2845.4 ± 206.3	11.0 ± 1.7a	344.2 ± 154.9	3916.0 ± 321.1a
**ABA**											
+ABA	196.1 ± 23.9	10.9 ± 1.3a	51.1 ± 3.8b	159.9 ± 11.8	2.0 ± 0.2	42.7 ± 2.3b	16.9 ± 1.4	2589.5 ± 233.3	9.7 ± 1.1	57.9 ± 21.1	3143.0 ± 269.9
−ABA	291.1 ± 54.7	15.4 ± 0.8b	72.8 ± 8.3a	266.0 ± 36.9	2.5 ± 0.4	73.9 ± 9.1a	18.4 ± 1.8	2410.3 ± 277.3	9.9 ± 1.7	314.0 ± 158.8	3480.8 ± 424.7
**WD**											
+WD	196.1 ± 51.7	14.0 ± 1.5	65.2 ± 9.6	161.8 ± 16.1	2.0 ± 0.4	45.2 ± 5.5	15.9 ± 1.6b	2036.6 ± 251.9b	7.5 ± 1.2b	27.1 ± 7.5	2606.2 ± 306.7b
−WD	291.1 ± 35.5	12.2 ± 0.9	58.7 ± 3.4	264.0 ± 35.7	2.5 ± 0.3	71.4 ± 8.4	19.5 ± 1.6a	2963.1 ± 174.1a	12.0 ± 1.4a	344.8 ± 154.9	4017.6 ± 271.7a
**+UV-B**											
+WD/+ABA	124.8 ± 21.3	13.8 ± 2.8	60.0 ± 3.9	158.4 ± 2.3	1.6 ± 0.4	37.5 ± 1.7	13.0 ± 1.9	1501.2 ± 81.4	8.3 ± 1.5	12.3 ± 4.9b	1934.8 ± 91.8
−WD/+ABA	170.4 ± 42.1	13.3 ± 0.3	55.0 ± 1.0	136.1 ± 9.8	2.2 ± 0.5	43.4 ± 2.4	18.4 ± 2.3	3045.7 ± 96.9	11.7 ± 1.9	24.9 ± 8.8b	3530.1 ± 135.2
+WD/−ABA	100.5 ± 6.9	16.8 ± 0.8	94.5 ± 30.3	182.5 ± 11.3	1.2 ± 0.6	52.9 ± 7.6	15.2 ± 3.5	1304.1 ± 38.4	4.5 ± 1.4	9.3 ± 2.5b	1787.0 ± 26.3
−WD/−ABA	286.3 ± 82.3	15.1 ± 1.5	55.9 ± 4.7	266.5 ± 51.1	1.7 ± 0.3	87.0 ± 13.6	18.8 ± 1.6	2766.5 ± 575.0	9.6 ± 2.0	64.1 ± 36.6b	3579.3 ± 699.8
**−UV-B**											
+WD/+ABA	225.4 ± 56.3	5.3 ± 1.5	37.9 ± 10.8	126.3 ± 18.1	1.4 ± 0.3	38.4 ± 5.2	15.3 ± 4.0	2725.3 ± 587.0	6.1 ± 0.6	29.9 ± 12.6b	3216.7 ± 690.3
−WD/+ABA	263.8 ± 40.1	11.1 ± 2.0	51.5 ± 6.6	218.6 ± 7.4	2.5 ± 0.5	51.3 ± 4.9	21.1 ± 1.5	3085.7 ± 91.9	12.5 ± 2.8	164.5 ± 43.3b	3890.4 ± 112.9
+WD/−ABA	459.4 ± 121.8	12.9 ± 1.3	68.4 ± 11.8	180.0 ± 65.1	3.6 ± 0.8	51.8 ± 21.8	20.2 ± 3.0	2615.9 ± 476.3	11.1 ± 4.1	56.8 ± 18.7b	3486.3 ± 670.2
−WD/−ABA	318.3 ± 107.3	16.7 ± 2.2	72.5 ± 7.5	434.9 ± 38.3	3.4 ± 0.7	104.0 ± 9.5	19.6 ± 6.5	2954.6 ± 536.6	14.3 ± 4.4	1125.7 ± 336.1a	5070.5 ± 571.0
**ANOVA ^a^**											
*p_(UV-B)_*	**0.0390**	0.0989	0.4278	**0.0224**	0.0680	0.1274	0.5693	0.1713	**0.0417**	0.0799	**0,0347**
*p_(ABA)_*	0.1414	**0.0011**	**0.0412**	**0.0019**	**0.0358**	**0.0018**	0.5506	0.5535	0.9051	**0.0121**	0,3722
*p_(WD)_*	0.4522	0.1481	0.4378	**0.0215**	0.3991	**0.0465**	**0.0334**	**0.0314**	**0.0169**	**0.0240**	**0,0231**
*p_(UV-B ×ABA)_*	0.4227	0.0501	0.6616	0.2527	**0.0013**	0.8029	0.9382	0.8438	0.1097	**0.0172**	0,3103
*p_(UV-B × WD)_*	0.0967	**0.0453**	0.1115	0.0627	0.9513	0.5320	0.4373	0.1130	0.8174	**0.0340**	0.5132
*p_(ABA × WD)_*	0.8702	0.3719	0.2621	**0.0208**	0.1368	**0.0391**	0.4259	0.9308	0.8402	**0.0152**	0.4607
*p_(UV-B ×ABA × WD)_*	0.2073	0.8242	0.5178	0.5641	0.2432	0.6967	0.6462	0.9598	0.5204	**0.0229**	0.6310

Values are means for each factor or treatment (n=3) ±SEM. Different lowercase letters within columns indicate significant differences (Fisher’s LSD, *p* ≤ 0.05). ^a^
*p*_(UV-B)_, *p*_(ABA)_ and *p*_(WD)_: effects of UV-B, ABA and WD, respectively; *p*_(UV-B ×ABA)_, *p*_(UV-B × WD)_, *p*_(ABA × WD)_, and *p*_(UV-B ×ABA × WD)_: interaction effects of factors. *p*-values ≤ 0.05 were considered statistically significant, and are in bold.

**Table 5 plants-10-00938-t005:** Volatile esters (µg L^−1^) determined in wines from Malbec plants exposed to combined UV-B, WD and ABA treatments.

Treatments	Ethyl Lactate	Ethyl Nonanoate	Ethyl Decanoate	Isoamyl Octanoate	Ethyl Succinate	Ethyl Dodecanoate	Ethyl Pentadecanoate	Total Esters
**UV-B**								
+UV-B	13.0 ± 3.3	3.3 ± 0.4	523.2 ± 65.9	47.4 ± 6.4	82.6 ± 11.5	11.7 ± 2.0	14.9 ± 4.6	7648.4 ± 928
−UV-B	8.1 ± 1.2	4.0 ± 0.5	655.5 ± 51.9	50.0 ± 5.4	106.5 ± 14.7	11.4 ± 2.2	20.7 ± 4.0	8802.3 ± 1197
**ABA**								
+ABA	7.4 ± 0.8b	3.0 ± 0.4b	547.3 ± 60.1	49.8 ± 5.9	91.5 ± 13.1	8.7 ± 2.0	21.7 ± 5.3	8150.1 ± 1018
−ABA	13.6 ± 3.3a	4.2 ± 0.4a	631.3 ± 62.4	47.6 ± 6.0	97.7 ± 14.2	14.5 ± 1.8	13.9 ± 2.9	8300.5 ± 1147
**WD**								
+WD	12.0 ± 3.5	3.4 ± 0.5	482.3 ± 63.8b	41.7 ± 6.1	84.9 ± 16.0	7.1 ± 1.1	8.7 ± 2.2b	7384.3 ± 1235
−WD	9.1 ± 1.0	3.8 ± 0.3	696.4 ± 40.9a	55.8 ± 5.0	104.3 ± 10.1	16.0 ± 2.1	26.9 ± 4.3a	9066.4 ± 836
**+UV-B**								
+WD/+ABA	10.0 ± 2.2	2.3 ± 0.7	363.6 ± 137.7	32.1 ± 13.8	40.5 ± 7.1	3.3 ± 1.0c	6.4 ± 5.1	5430.6 ± 1633
−WD/+ABA	6.3 ± 0.6	3.1 ± 0.4	721.0 ± 84.6	67.3 ± 10.0	111.8 ± 15.6	17.8 ± 5.0ab	34.1 ± 12.1	10,013.3 ± 1264
+WD/−ABA	26.7 ± 9.9	3.4 ± 0.8	337.4 ± 19.5	29.7 ± 5.3	54.2 ± 3.9	10.5 ± 0.9bc	3.7 ± 0.4	4941.1 ± 186
−WD/−ABA	8.9 ± 2.1	4.2 ± 0.8	670.8 ± 93.1	60.6 ± 5.4	124.0 ± 5.3	15.2 ± 1.9ab	15.4 ± 3.4	10,208.4 ± 1564
**−UV-B**								
+WD/+ABA	5.4 ± 1.0	2.8 ± 1.1	461.0 ± 95.9	41.9 ± 7.9	118.8 ± 39.6	5.1 ± 1.5c	10.4 ± 3.9	8418.0 ± 3455
−WD/+ABA	8.1 ± 1.4	3.8 ± 0.9	643.8 ± 62.1	58.0 ± 6.2	94.7 ± 7.3	8.4 ± 0.4bc	35.9 ± 7.8	8738.7 ± 973
+WD/−ABA	5.9 ± 3.0	5.0 ± 0.9	767.3 ± 52.6	62.9 ± 13.4	125.9 ± 33.8	9.6 ± 1.8bc	14.3 ± 6.2	10,747.5 ± 2797
−WD/−ABA	13.0 ± 1.9	4.2 ± 0.7	749.9 ± 117.0	37.3 ± 11.7	86.6 ± 38.5	22.7 ± 3.4a	22.2 ± 6.9	7305.2 ± 2775
**ANOVA ^a^**								
*p_(UV-B)_*	0.1625	0.5053	0.2595	0.8482	0.5150	0.8549	0.2368	0.6694
*p_(ABA)_*	**0.0293**	**0.0066**	0.2383	0.7282	0.5161	**0.0250**	0.2135	0.8789
*p_(WD)_*	0.3428	0.5813	**0.0147**	0.0944	0.1480	**0.0019**	**0.0013**	0.2472
*p_(UV-B ×ABA)_*	0.1768	0.7491	0.1009	0.7045	0.4830	0.1303	0.6308	0.7636
*p_(UV-B × WD)_*	**0.0454**	0.6608	0.0651	**0.0433**	**0.0092**	0.6123	0.5402	**0.0490**
*p_(ABA × WD)_*	0.3320	0.2474	0.4197	0.0933	0.6619	0.9908	0.1836	0.4442
*p_(UV-B ×ABA × WD)_*	0.0822	0.2302	0.5226	0.1613	0.7167	**0.0491**	0.9487	0.2785

Values are means for each factor or treatment (n=3) ±SEM. Different lowercase letters within columns indicate significant differences (Fisher’s LSD, *p* ≤ 0.05). ^a^
*p*_(UV-B)_, *p*_(ABA)_ and *p*_(WD)_: effects of UV-B, ABA and WD, respectively; *p*_(UV-B ×ABA)_, *p*_(UV-B × WD)_, *p*_(ABA × WD)_, and *p*_(UV-B ×ABA × WD)_: interaction effects of factors. *p*-values ≤ 0.05 were considered statistically significant, and are in bold.

**Table 6 plants-10-00938-t006:** Acetoin and volatile organic acid (µg L^−1^) determined in wines from Malbec plants exposed to combined UV-B, WD and ABA treatments.

Treatments	Acetoin	Hexanoic Acid	Octanoic Acid	Palmitic Acid
**UV-B**				
+UV-B	9.9 ± 1.3	9.8 ± 1.8	24.4 ± 3.6	0.4 ± 0.4b
−UV-B	9.4 ± 2.4	20.3 ± 3.7	28.4 ± 4.1	5.0 ± 1.3a
**ABA**				
+ABA	6.6 ± 1.0	17.1 ± 3.6	29.3 ± 4.0	1.5 ± 0.9
−ABA	12.8 ± 2.1	13.0 ± 2.9	23.5 ± 3.6	3.8 ± 1.4
**WD**				
+WD	7.6 ± 1.6	12.7 ± 1.6	18.2 ± 1.8b	1.4 ± 1.2
−WD	11.8 ± 2.0	17.5 ± 4.3	34.6 ± 3.9a	4.0 ± 1.1
**+UV-B**				
+WD/+ABA	5.9 ± 2.8bc	18.7 ± 1.5	17.6 ± 4.7	0.00 ± 0.0
−WD/+ABA	9.7 ± 1.7bc	7.1 ± 1.0	41.1 ± 3.8	0.00 ± 0.0
+WD/−ABA	14.1 ± 2.7ab	9.1 ± 1.9	15.7 ± 4.3	0.00 ± 0.0
−WD/−ABA	10.0 ± 0.5bc	4.4 ± 2.5	23.2 ± 5.7	1.47 ± 1.5
**−UV-B**				
+WD/+ABA	3.7 ± 1.4c	7.4 ± 0.9	17.2 ± 2.2	0.00 ± 0.0
−WD/+ABA	6.9 ± 0.9bc	35.3 ± 4.3	41.4 ± 4.6	6.17 ± 1.6
+WD/−ABA	6.5 ± 2.8bc	15.5 ± 2.1	22.2 ± 3.2	5.63 ± 4.3
−WD/−ABA	20.4 ± 5.7a	23.2 ± 8.5	32.8 ± 12.3	8.27 ± 0.5
**ANOVA ^a^**				
*p_(UV-B)_*	0.8529	**0.0100**	0.3498	0.0571
*p_(ABA)_*	**0.0142**	0.2477	0.2276	0.0850
*p_(WD)_*	0.0631	0.0962	**0.0231**	0.1645
*p_(UV-B ×ABA)_*	0.3508	0.5440	0.3895	0.2175
*p_(UV-B × WD)_*	0.0567	**0.0043**	0.8461	0.2917
*p_(ABA × WD)_*	0.7395	0.3389	0.1350	0.6706
*p_(UV-B ×ABA × WD)_*	**0.0460**	0.0711	0.8961	0.3167

Values are means for each factor or treatment (n=3) ±SEM. Different lowercase letters within columns indicate significant differences (Fisher’s LSD, *p* ≤ 0.05). ^a^
*p*_(UV-B)_, *p*_(ABA)_ and *p*_(WD)_: effects of UV-B, ABA and WD, respectively; *p*_(UV-B ×ABA)_, *p*_(UV-B × WD)_, *p*_(ABA × WD)_, and *p*_(UV-B ×ABA × WD)_: interaction effects of factors. *p*-values ≤ 0.05 were considered statistically significant, and are in bold.

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
