# Peer review of "Abscisic Acid’s Role in the Modulation of Compounds that Contribute to Wine Quality"

_plants, 2021, doi:10.3390/plants10050938_

Round 1

Reviewer 1 Report

The manuscript “Abscisic acid role in the modulation of compounds that contribute to the wine quality” is a very interesting contribution to viticulture. However, there are some important considerations that are not clear to me after reading this manuscript. How did the authors relate the application rate of ABA to the grapevines with the phenolic and volatile compound content of the wine? Even though the concentration of phenolic compounds in wines depends on the degree of the extraction of these compounds from grapes, the content volatile compounds in wines depends on the yeast activity during fermentation regarding sugars and amino acids. In this way, little information was provided about grape yield, amino acids precursors of volatile compounds or other components that may bring more information about the results exposed. The authors mention in the introduction that volatile compounds have been less studied but the volatiles that they studied were to a greater extent those produced after alcoholic fermentation. On the other hand, I think it is important to compare the application of ABA to the differences with or without UV-B and with stress independently to see if there are differences in combination.

Some minor considerations are described in the following lines:

Abstract

L20-21. The grammar of this sentence could be improved

L24-25. This sentence could be deleted

Introduction

I think that during the introduction it is necessary to explain the differences between the volatiles of the grape and those of the wine, including that some of the grape evolve from its redox in the aging of the wine

L57-58. To my opinion, this is because the analysis of VOC's in grapes is complex and deserves more research

Materials and Methods

L126. Please, explain deeply the procedures of phenolic compounds analysis in wines

L136. Please, explain deeply the analysis of volatile compounds in wines

Discussion

I think it is important to know if ABA is more effective under UVB+ than UVB- conditions. I suggest to discuss it in relation to the differences on the results and the exposed literature

L290-292. It could be attributed to the analytical technique or the time the sample was frozen

L297-299. What about the increase in skin pulp ratio? There is no information about grape yield in the exposed results.

L302-304. This sentence is not related to the results exposed and could be deleted

L307-309. It is important to define the level of stress applied by water deficit and also, the doses of ABA applications to grapevines

L311. I think it is important to define the level of stress. I suggest to add some information related to water potential or stomatal conductance

L370-371. I think it is important to define the level of stress. I suggest to add some information related to water potential or stomatal conductance

Conclusions

This section should be improved according to the results exposed

Author Response

The manuscript “Abscisic acid role in the modulation of compounds that contribute to the wine quality” is a very interesting contribution to viticulture. However, there are some important considerations that are not clear to me after reading this manuscript.

Q1: How did the authors relate the application rate of ABA to the grapevines with the phenolic and volatile compound content of the wine? Even though the concentration of phenolic compounds in wines depends on the degree of the extraction of these compounds from grapes, the content volatile compounds in wines depends on the yeast activity during fermentation regarding sugars and amino acids. In this way, little information was provided about grape yield, amino acids precursors of volatile compounds or other components that may bring more information about the results exposed.

R1: We agree with the reviewer that little information is provided about amino acids or other various compounds as precursors of VOCs in fruits. However, to accomplish such task might be overwhelming in both senses, work to be done and data presented, and therefore exceed the scope of the article. So considering that their content in wines depends on the yeast activity during fermentation, the wines were elaborated by a standardized procedure, by inoculation with a selected commercial Saccharomyces cerevisiae bayanus yeast. Besides, the effect of treatments in fruit yield and in the composition of the berries was previously reported and discussed in Alonso et al., 2016. We have now included a brief mention related with fruit yield in the discussion section that reads: “In Alonso et al. (2016) we found that fruit yield was reduced in +UV-B/+D and in +ABA/+D treatments, i.e., WD only affected yield in plants under high intensity of UV-B or in combination with ABA applications. In addition, ABA sprays may anticipate the accumulation of sugars in berries, but major effects of ABA are found at veraison (phenological effect) and differences are reduced at harvest [21].”

Q2: The authors mention in the introduction that volatile compounds have been less studied but the volatiles that they studied were to a greater extent those produced after alcoholic fermentation. On the other hand, I think it is important to compare the application of ABA to the differences with or without UV-B and with stress independently to see if there are differences in combination.

R2: The reviewer’s concern may be satisfied if we consider that the effects of a single factor, i.e. UV-B (PUV-B), ABA (PABA) and water restriction (PD), and the effects of their interactions are both presented, allowing to evaluate their effects alone or in combination.

Some minor considerations are described in the following lines:

 Abstract

Q3: L20-21. The grammar of this sentence could be improved

R3: The sentence in the abstract was modified with the intention of improving it: “Two UV-B treatments (ambient solar UV-B or reduced UV-B), two watering treatments (well-watered or moderate water deficit) and two ABA treatments (no ABA and sprayed ABA) were given in a factorial design, during one growing season”.

Q4: L24-25. This sentence could be deleted

R4: Deleted as suggested.

Introduction

Q5: I think that during the introduction it is necessary to explain the differences between the volatiles of the grape and those of the wine, including that some of the grape evolve from its redox in the aging of the wine

R5: Thanks for the suggestion. It was added “The aroma of wine is one of the main factors that determines its quality, and derives from both the VOCs present in the berries (Canuti et al., 2009) and those produced during the fermentation and aging (Bartowsky and Pretorius, 2009; Garde-Cerdán et al., 2010)”

Q6: L57-58. To my opinion, this is because the analysis of VOC's in grapes is complex and deserves more research

R6: We agree with the reviewer´s comment.

Materials and Methods

Q7: L126. Please, explain deeply the procedures of phenolic compounds analysis in wines

R7: The procedure details are now included in the M&M section:

”LMWPs and anthocyanins were assessed in August 2013, using a high performance liquid chromatography-multiple wavelength detector (HPLC-MWD; Dionex Ultimate 3000, Dionex Softron GmbH, Thermo Fisher Scientific Inc., Germering, Germany) following the procedures described by Fontana et al. (2016). For LMWPs, a 5 ml aliquot of wine was acidified with 57 µl of 1% FA. Then 2.5 mL of MeCN was added and it was stirred vigorously for 60 sec. Later, 4 g of Na2SO4 and 1.5 g of NaCl were added. The mixture was stirred again for 60 sec and centrifuged for 10 min at 300 rpm. An aliquot of 1 ml was extracted from the MeCN phase and 150 mg of CaCl2, 100 mg of PSA and 100 mg of C18 were added. The mixture was vortexed for 30 sec and centrifuged 2.5 min at 12000 rpm. Finally, a 500 µl aliquot of the supernatant was transferred to a vial and evaporated to dryness with a stream of N2. The residue was resuspended with 500 µl of the initial mobile phase and injected into the HPLC-MWD. HPLC separations were carried out in reversed-phase Kinetex C18 column (3.0 mm x 100 mm, 2.6 µm) Phenomenex (Torrance, CA, USA). Ultrapure water with 0.1% FA (A) and MeCN (B) were used as mobile phases. Analytes were separated using the following gradient: 0–2.7 min, 5% B; 2.7–11 min, 30% B; 11–14 min, 95% B; 14–15.5 min, 95% B; 15.5–17 min, 5% B: 17–20, 5% B. The mobile phase flow was 0.8 ml min-1. The column temperature was 35 °C and the injection volume 5 µL. The identification and quantification were carried out by comparing the retention time and the maximum absorbance value detected in a peak of the sample with those obtained in the injection of the pure standards.

For anthocyanins, HPLC separations were carried out in reversed-phase C18 column (3.0 mm x 100 mm, 2.6 µm, Kinetex, Phenomenex; Torrance, CA, USA). The mobile phase consisted of ultra-pure water/ FA/ MeCN (87: 10: 3, v/v/v; eluent A) and ultra-pure water/FA/MeCN (40:10:50, v/v/v; eluent B), using the following gradient: 0 min, 10% B; 0-6 min, 25% B; 6-10 min, 31% B; 10-11min, 40% B; 11-14 min, 50% B; 14-15 min, 100% B; 15-17 min, 10% B; 17-21 min, 10% B. The mobile phase flow was 1 ml min-1, the column temperature was kept at 25 °C and the injection volume was 5 µl. Quantification was car-ried out by measuring the area under the peaks at 520 nm, and the anthocyanin content was expressed as malvidin-3-glucoside, using a calibration curve with an external standard (1-250 µg ml-1, R2 = 0.9984).

The Chromeleon 7.1 software was used to control all the acquisition parameters of the LC-MWD system and also to process the obtained data.”

Q8: L136. Please, explain deeply the analysis of volatile compounds in wines

R8: The procedure details are now included in the M&M section:

“The wine samples were analyzed in August 3013, by headspace solid-phase micro-extraction (HS-SPME), following the procedures described by Xi et al. (2011) with some modifications: 8 ml of wine and 20 µl of 2-octanol (1:2000; internal standard) were transferred into a 20 ml vial, and 1.5 g of NaCl added. The vial was sealed with a Teflon septum and equilibrated at 40 ºC for 10 min, shaken by a magnetic stirrer (540 rpm). The fiber PDMS (100 mm polydimethylsiloxane, Supelco, Bellefonte, PA, USA) was inserted into the headspace, where extraction was allowed to occur for 30 min with continued heating (40 °C) and shaking. This was immediately followed by desorption of the analytes into the gas chromatography-electron impact mass spectrometer (GC-MS; Clarus 500, PerkinElmer, Shelton, CT, USA). Separation of compounds was performed with an Agilent DB-Wax capillary column (30 m x 0.32 mm, 0.25 μm) using helium as carrier gas at a flow rate of 1 ml min-1. The oven temperature program was as follows: 40 °C for 3 min, then it was in-creased to 160 °C at 5 °C min-1, then it was raised to 230 °C at 7 °C min-1 and finally it was maintained 8 min at 230 °C. The mass spectrometer was operated using an impact energy of 70 eV and scanned in a range of 35 to 200 atomic mass units. The metabolites were identified by comparison of GC retention times and full mass spectra of the corresponding standard obtained by Fluka (Sigma-Aldrich Steinheim, Switzerland) and, by using their fragmentation pattern and comparison with data of the NIST library. Peak areas were referred to the standard 2-octanol for quantification.” 

Discussion

Q9: I think it is important to know if ABA is more effective under UVB+ than UVB- conditions. I suggest to discuss it in relation to the differences on the results and the exposed literature

R9: A paragraph was added in the discussion that now reads “Although ABA applications increased certain compounds regardless of the intensity of UV-B received, the greater effectiveness of +ABA was observed in plants under high UV-B irradiances, since those compounds with high antioxidant capacity, such as piceid, and those related to oenological quality, such as peonidin and delphinidin, were markedly increased (as compared to -UV-B treatment). Similar results were observed in a previous work, where ABA applications in plants under +UV-B produced the highest berry skin ABA levels, increasing additively flavonols and anthocyanins (Berli et al., 2011).”

Q10: L290-292. It could be attributed to the analytical technique or the time the sample was frozen

R10: Sorry for the confusion. It is now clearly stated that in the present work, the HPLC analysis of the bottled wines was performed during August 2013, i.e. only a few months after the winemaking process.

Q11: L297-299. What about the increase in skin pulp ratio? There is no information about grape yield in the exposed results.

R11: A brief mention was made based on the results of Alonso et al. (2016). Now reads “In Alonso et al. (2016) we found that fruit yield was reduced in +UV-B/+D treatments and in +ABA/+D treatments, i.e., WD only affected yield in plants under high intensity of UV-B or in combination with ABA applications.”

Q12: L302-304. This sentence is not related to the results exposed and could be deleted

R12: The sentence was deleted as suggested.

Q13: L307-309. It is important to define the level of stress applied by water deficit and also, the doses of ABA applications to grapevines

R13:Now reads “Deis et al. (2011) found in cv. Cabernet Sauvignon that post-veraison WD (stem water potential between -1.0 and -0.5 MPa, moderate intensity according to Van Leeuwen et al., 2009) and applied ABA (doses of 250 mg l-1) increased (+)-catechin in berries and wines, while Koundouras et al. (2013) did not observe an effect of post-veraison WD (stem water potential between -1.2 and -0.8 MPa) in cv. Agiorgitiko berries or wine.”

Q14: L311. I think it is important to define the level of stress. I suggest to add some information related to water potential or stomatal conductance

R14: Now reads “The biosynthesis of stilbenes, including trans-resveratrol, is increased in berries by environmental signals such as WD (stem water potentials between -1.25 and -0.8MPa; Deluc et al., 2011)”

Q15: L370-371. I think it is important to define the level of stress. I suggest to add some information related to water potential or stomatal conductance

R15: Now reads “Malbec belongs to the group of varieties with very low concentration of terpenoids and possibly explain the discrepancies of our results with those of Ou et al. (2010), whose midday leaf water potential of WD treatments were between -1.09 and -1.63 (moderate to severe intensity according to Van Leeuwen et al., 2009)

Conclusions

Q16: This section should be improved according to the results exposed

R16: Conclusions were shortened and expressed more clearly according to the results.

Reviewer 2 Report

The manuscript entitled:  “Abscisic acid role in the modulation of compounds that contribute to the wine quality“- reports a study on the different growth factors on volatile organic compounds (VOCs), polyphenols, and antioxidant activity.

Abstract: Authors wrote: "Under these treatments, alcohols, VOCs and esters are scarcely affected." alcohols and esters are VOCs, this sentence should be rewritten.

The antioxidant activity is not mentioned in the abstract.

"The metabolic, physiological and technical mechanisms behind the ABA role in the wines obtained are discussed in confront with the existent literature"- these sentence is very general, and it is not a proper conclusion.

The introduction is well written, however, "most literature has been oriented to study VOC’s as a product of must fermentation (Berbegal et al., 2020)" and only one literature position is cited. The Authors have clearly specified the purpose of their work.

The methods used seem appropriate to ensure a solid experimental design, but there are some small flaws. For example, any of the standards that were used in the experiments were mentioned in "Materials". in the title of the subchapter there should not be a dot e.g. "2.5. Analysis of wine VOCs." There is no information about the reference to VOCs extraction procedure. Moreover, no information about DB-Wax column parameters (length, particle size). 

The analysis and interpretation of the results are adequate.

The Tables are not readable. Please work more on the visualization of Tables. The total sum of compounds should be in the last column.

Most of this manuscript is tables and figures, not much text. No ORAC results presented, neither described and discussed.

References are appropriate but they are not written in the style of this Journal neither in "References" nor in text. 

It was hard to decide because the quality of the manuscript is not good, however, if the Authors work hard on the text once more, I give it a chance and recommend a major revision.

Author Response

The manuscript entitled:  “Abscisic acid role in the modulation of compounds that contribute to the wine quality“- reports a study on the different growth factors on volatile organic compounds (VOCs), polyphenols, and antioxidant activity.

Q1: Abstract: Authors wrote: "Under these treatments, alcohols, VOCs and esters are scarcely affected." alcohols and esters are VOCs, this sentence should be rewritten.

R1: The words “alcohols” and “esters” were deleted. Now reads: “Under these treatments, VOCs were scarcely affected.”

Q2: The antioxidant activity is not mentioned in the abstract.

R2: The abstract now reads: “Volatile organic compounds (VOCs) and phenolic compounds (PCs) accumulate in the berry skins, possess antioxidant activity and are important attributes for red wine.

Q3: "The metabolic, physiological and technical mechanisms behind the ABA role in the wines obtained are discussed in confront with the existent literature"- these sentence is very general, and it is not a proper conclusion.

R3: As suggested also by reviewer#1 the sentence was deleted.

Q4: The introduction is well written, however, "most literature has been oriented to study VOC’s as a product of must fermentation (Berbegal et al., 2020)" and only one literature position is cited. 

R4: A complementary citation was added and reads: “most literature has been oriented to study VOC’s as a product of must fermentation (Berbegal et al., 2020; Capone et al., 2021),”

The Authors have clearly specified the purpose of their work.

The methods used seem appropriate to ensure a solid experimental design, but there are some small flaws.

Q5: For example, any of the standards that were used in the experiments were mentioned in "Materials".

R5: A section of standards was added in M&M section.

Q6:  in the title of the subchapter there should not be a dot e.g. "2.5. Analysis of wine VOCs." There is no information about the reference to VOCs extraction procedure. Moreover, no information about DB-Wax column parameters (length, particle size).

R6: Thanks, the dot was deleted a reference to the VOCs extraction procedure was added, and a description about the column was included.

The analysis and interpretation of the results are adequate.

Q7: The Tables are not readable. Please work more on the visualization of Tables. The total sum of compounds should be in the last column.

R7: The size of the tables were reduced, deleting those compounds that were not affected by the treatments (P≥0.05). In Table 1, gallic acid, catechin, epicatechin, gallocatechin and tyrosol were deleted. Also, farnesol (in Table 1) and isoamyl acetate, ethyl octanoate, octyl formate and ethyl palmitate (in Table 5) were deleted. Finally, the total sum of the compounds was placed in the last column.

Q8: Most of this manuscript is tables and figures, not much text. No ORAC results presented, neither described and discussed.

R8: ORAC results are presented in the  Results section: “Respect to ORAC, the -D/+UV-B wines presented a higher ORAC as compared to -D/-UV-B wines (Figure 5.B)”. The discussion section now reads “Respect to antioxidant capacity, it was observed that only the +UV-B /-D wines presented the highest values, measuring by the ORAC technique. According to Rodríguez-Bonilla et al. (2017), possibly, the use of other complementary technique would be necessary, since we observed that UV-B increased the LMWP with highest antioxidants capacity, regardless of the water status.”

Q9: References are appropriate but they are not written in the style of this Journal neither in "References" nor in text. 

R9: Now the references are correctly written as required by the Journal.

Q10: It was hard to decide because the quality of the manuscript is not good, however, if the Authors work hard on the text once more, I give it a chance and recommend a major revision.

R10: We are grateful for the opportunity and hope that the revised version of the manuscript fulfills the requirements needed to be published.

Reviewer 3 Report

Dear Authors/Editor,

The aim of the article „Abscisic acid role in the modulation of compounds that contribute to the wine quality“ was to analyze the ABA role in the modulation of compounds that contribute to the wine quality, by comparing the independent and interactive effects of contrasting UV-B levels, WD and sprayed ABA, on PCs and VOCs profiles of wines. Volatile organic compounds (VOCs) and phenolic compounds (PCs) are considered the most important attributes for wine quality, although most scientific works evaluate the interactive effects of environmental factor on these compounds in berries, but not in the wine. According to the authors, there are no reports of VOCs profiles in wine from berries of vines treated with ABA. Therefore, the research described in the manuscript is relevant and important both scientifically and practically. The manuscript is written consistently and comprehensibly enough.

A few remarks:

The manuscript was not prepared in accordance with the requirements –Plants journal Instructions for authors, including manuscript section order (Materials and methods should be after the Discussion) and References (references must be numbered in order of appearance in the text and listed individually at the end of the manuscript).

Line 31-33. The sentence “Also, grape secondary metabolism was linked to abiotic stress, especially to the ABA-mediated responses“ - a self-evident statement common to all plants, not just grapes.

Line 41- 45. It would be good to provide references to literature, for example: Šikuten et al, 2020 (https://doi.org/10.3390/molecules25235604) or Simonetti et al, 2020 (https://doi.org/10.3390/molecules25163748)

Line 91. The experiments described in the manuscript was performed in 2013 and some results from this treatment is published by the authors in 2015-2016. Why such a long time lag between grape picking and submission of wine analysis data? Is it related to the production and preservation of wine? The authors do not provide clear information on wine preserving duration and time of wine composition analysis.

Line 94. In the text: “Briefly, a low UV-B treatment (-UV-B) was set by using a polyester cover that absorbed 78% of UV-B and 18% of ultraviolet-A (UV-A) radiation”. Didn’t this cover reduce not only of the UV-B but also light intensity of other wavelengths? How could the overall decrease in light intensity affect the study results?

Line 191. Table 1 as well as tables 4 and 5 may be too large and difficult to format. It might be advisable to present these tables in the Supplementary Materials.

The fonts of the tables should be harmonized

More references less than 5 years old could be included in the introduction and discussion.

Author Response

The aim of the article „Abscisic acid role in the modulation of compounds that contribute to the wine quality“ was to analyze the ABA role in the modulation of compounds that contribute to the wine quality, by comparing the independent and interactive effects of contrasting UV-B levels, WD and sprayed ABA, on PCs and VOCs profiles of wines. Volatile organic compounds (VOCs) and phenolic compounds (PCs) are considered the most important attributes for wine quality, although most scientific works evaluate the interactive effects of environmental factor on these compounds in berries, but not in the wine. According to the authors, there are no reports of VOCs profiles in wine from berries of vines treated with ABA. Therefore, the research described in the manuscript is relevant and important both scientifically and practically. The manuscript is written consistently and comprehensibly enough.

A few remarks:

Q1: The manuscript was not prepared in accordance with the requirements –Plants journal Instructions for authors, including manuscript section order (Materials and methods should be after the Discussion) and References (references must be numbered in order of appearance in the text and listed individually at the end of the manuscript).

R1: We appreciate the evaluation of the work, highlighting its relevance. Now, the Journal Instruction for Authors were taken into account and M&M section is after the Discussion and the References are correctly written.

Q2: Line 31-33. The sentence “Also, grape secondary metabolism was linked to abiotic stress, especially to the ABA-mediated responses“ - a self-evident statement common to all plants, not just grapes.

R2: We agree with the reviewer´s suggestion and the word “grape” was deleted.

Q3: Line 41- 45. It would be good to provide references to literature, for example: Šikuten et al, 2020 (https://doi.org/10.3390/molecules25235604) or Simonetti et al, 2020 (https://doi.org/10.3390/molecules25163748)

R3: Šikuten et al. 2020 was added as a reference for the sentence.  

Q4: Line 91. The experiments described in the manuscript was performed in 2013 and some results from this treatment is published by the authors in 2015-2016. Why such a long time lag between grape picking and submission of wine analysis data? Is it related to the production and preservation of wine? The authors do not provide clear information on wine preserving duration and time of wine composition analysis.

R4: Sorry for the confusion. It is now clearly stated that the analysis of the wine was performed during August 2013, i.e, few months after the winemaking process.

Q5: Line 94. In the text: “Briefly, a low UV-B treatment (-UV-B) was set by using a polyester cover that absorbed 78% of UV-B and 18% of ultraviolet-A (UV-A) radiation”. Didn’t this cover reduce not only of the UV-B but also light intensity of other wavelengths? How could the overall decrease in light intensity affect the study results?

R5: In a previous paper, Alonso et al., 2015 a major description of the experimental design is presented. Additionally, in Berli et al., 2008 the transmittance spectral characteristics of the covers is published. The plastic cover used for -UV-B , also affected  the solar UV-A (18%) and PAR (12%), but the percentages of absorption are similar of those of the polyethylene used for +UV-B treatment.  Therefore, there is no significant change in solar light quality, other than UV-B.

Q6: Line 191. Table 1 as well as tables 4 and 5 may be too large and difficult to format. It might be advisable to present these tables in the Supplementary Materials.

R6: The size of the Tables were reduced as suggested by reviewer#2, deleting those compounds that were not affected by the treatments (P≥0.05).

Q7: The fonts of the tables should be harmonized

R7: Corrected as suggested

Q8: More references less than 5 years old could be included in the introduction and discussion.

R8: In the discussion was added Rodríguez-Bonilla et al. (2017), and in the introduction Capone et al. (2021).    

Round 2

Reviewer 1 Report

I am totally satisfied with the answer and I congratulate the authors for their effort. There are only a few formatting errors

Another minor revision.
Line 408. Please, add the unit (MPa) to values

Reviewer 2 Report

Thanks to Authors for their work to improve their manuscript. I recommend to publish this manuscript in the present form.